# Out-of-distribution Few-shot Learning For Edge Devices without Model Fine-tuning

## Abstract

Few-shot learning (FSL) via customization of a deep learning network with limited data has emerged as a promising technique to achieve personalized user experiences on edge devices. However, existing FSL methods primarily assume independent and identically distributed (IID) data and utilize either computational backpropagation updates for each task or a common model with task-specific prototypes. Unfortunately, the former solution is infeasible for edge devices that lack on-device backpropagation capabilities, while the latter often struggles with limited generalization ability, especially for out-of-distribution (OOD) data. This paper proposes a lightweight, plug-and-play FSL module called Task-aware Normalization (TANO) that enables efficient and task-aware adaptation of a deep neural network without backpropagation. TANO covers the properties of multiple user groups by coordinating the updates of several groups of the normalization statistics during meta-training and automatically identifies the appropriate normalization group for a downstream few-shot task. Consequently, TANO provides stable but task-specific estimations of the normalization statistics to close the distribution gaps and achieve efficient model adaptation. Results on both intra-domain and out-of-domain generalization experiments demonstrate that TANO outperforms recent methods in terms of accuracy, inference speed, and model size. Moreover, TANO achieves promising results on widely-used FSL benchmarks and data from real applications.

## 1 Introduction

Due to the rapid growth of mobile hardware and software, deep neural networks (DNNs) have become increasingly prevalent on edge devices, including mobile phones, tablet computers, and smart watches, to provide users with various services, such as news recommendation, face recognition, and photo beautification (Deng et al., 2019; Cui et al., 2020; Chang et al., 2020). Typically, a common model trained on the server is utilized for different users (Lange et al., 2020). However, as user demands continue to become more diverse, personalized services are desired to enhance user experiences. To this end, various algorithms have been proposed to adapt DNNs to a user's own data and provide personalized services (Yu et al., 2019; Chen et al., 2021; Chaudhuri et al., 2020). Nonetheless, adapting DNNs on edge devices faces two significant challenges: limited computational resources and memory space and the few-shot learning (FSL) problem arising from the small data size of a user.

To address the FSL challenge, meta-learning methods have been developed to train a deep learning model capable of adapting to new tasks with few samples without overfitting (Sun et al., 2019; Liu et al., 2019; Yan et al., 2021). In meta-learning, a model with fast adaptation ability is trained using vast auxiliary data in the meta-training stage, and then adapted to a target task with limited data in the meta-testing stage. Existing FSL meta-learning methods can be categorized as gradient-based or metric-based algorithms. Gradient-based algorithms train a model with adaptation ability in the meta-training stage that adapts to a specific task via several steps of gradient descent in the meta-testing stage (Finn et al., 2017; Vuorio et al., 2019; Rajeswaran et al., 2019). In contrast, metric-based methods learn a common feature extractor to encode all data and make predictions via a similarity-based metric function (Snell et al., 2017; Vinyals et al.,

2016; Sung et al., 2018; Lee et al., 2019). Despite the plethora of FSL algorithms, lightweight adaptation remains a challenge, which is the second challenge faced when adapting DNNs on edge devices.

For mobile devices, backpropagation-based adaptation algorithms (Finn et al., 2017; Vuorio et al., 2019; Rajeswaran et al., 2019) are not practical for on-device user-specific adaptation, as common deep learning platforms like Tensorflow Lite or PyTorch Mobile only support model inference on devices, not model training. Metric-based methods circumvent backpropagation during testing (Snell et al., 2017; Vinyals et al., 2016; Sung et al., 2018; Lee et al., 2019). However, a fixed model may not capture the varying preferences of different users, particularly in out-of-distribution (OOD) scenarios where data come from different distributions (see Table 2 for some evidences). It is still challenging to develop a lightweight FSL method for OOD data.

In this paper, we consider three different FSL scenarios, including (1) intra-domain FSL, (2) out-of-domain FSL, and standard FSL. In intra-domain FSL, the support set consists of data from multiple domains with different data distributions, and the query set contains data from these seen domains with unseen classes. In out-of-domain FSL, the query set comprises data from domains unseen during meta-training. In standard FSL, both the support set and query set come from a single domain with independent and identically distributed (IID) data. See Figure 1 as an illustration. One practical example for is the face identification in mobile phones, where only a few photos are available for each user, making it a few-shot learning problem. In industry, this task may involve training and testing data from one user, the same group of users, or different groups of users, corresponding to standard FSL, intra-domain FSL, and out-of-domain FSL, respectively.

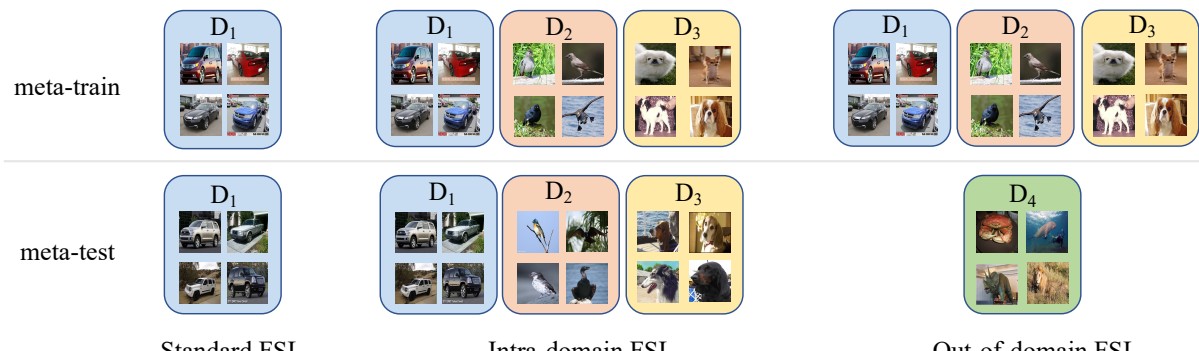

Figure 1: **Few-shot learning settings.** This figure illustrates three few-shot learning (FSL) settings: standard FSL (left), intra-domain FSL (middle), and out-of-domain FSL (right). In standard FSL, both meta-train and meta-test data come from a single domain. In intra-domain FSL, both meta-train and meta-test data come from the same group of domains. In out-of-domain FSL, meta-train and meta-test data come from different domains.

To address the out-of-distrubution FSL problem, the paper proposes Task-aware Batch Normalization (TANO), a lightweight, plug-and-play FSL module for standard meta-learning algorithms. The intuition behind TANO is to learn a task-specific estimation of Batch Normalization (BN) statistics in a deep neural network (DNN) to adjust the model according to the input data's distribution. BN aligns feature maps to a standard distribution, which reduces domain bias, and its parameters contain domain-specific information. AdaptiveBN (Li et al., 2018) and DSBN (Chang et al., 2019) have shown the effectiveness of BN modification for domain adaptation without model training. However, obtaining an accurate estimation of BN statistics for a target task on-device is challenging in FSL settings due to the limited data size. To overcome this limitation, the paper proposes Group Workers consisting of multiple groups of BN layers to encode domain-specific features during the meta-training stage, along with a Group Coordinator that learns to coordinate different BN groups over few-shot tasks (see Figure 2). TANO can be simultaneously learned with metric-based FSL methods. Once trained, TANO can handle intra-domain and out-of-domain FSL tasks by providing task-specific but stable estimation of BN statistics to close the distribution gaps among data. Furthermore, TANO can significantly enhance the performance of vanilla meta-learning algorithms in

standard FSL benchmarks. In summary, TANO provides a practical solution to on-device FSL with variable data and limited computational ability. The main contributions of this paper are three-fold:

1. We propose TANO, a lightweight module for metric-based meta-learning algorithms that significantly boost model performance in FSL with data from multiple domains *without gradient updates.*

2. Experimental results show that TANO outperforms recent methods for FSL with data from multiple domains. It also shows satisfactory performance for out-of-domain generalization in FSL.

3. TANO significantly boost performance of vanilla FSL algorithms in standard FSL benchmarks, including *mini*ImageNet and *tiered*ImageNet.

## 2 Related Work

**Few-shot Learning.**  Recent progress in meta-learning for few-shot learning (FSL) includes gradient-based and metric-based approaches (Snell et al., 2017; Sung et al., 2018; Lee et al., 2019; Xu et al., 2021). Gradient-based methods (Vuorio et al., 2019; Rusu et al., 2019; Rajeswaran et al., 2019; Oh et al., 2021; Snell & Zemel, 2021; Afrasiyabi et al., 2020; Ravi & Larochelle, 2017) learn a model that can be fine-tuned for specific tasks with a few gradient updates. Specifically, Vuorio et al. (2019) and Ravi & Larochelle (2017) assumed that a good model initialization on the base classes, which is learned in a meta-learning manner, can also lead to rapid convergence on the novel classes. Rusu et al. (2019) learns a task-dependent initialization in a latent generative representation space. Rajeswaran et al. (2019) decouples the meta-gradient computation from the inner loop optimizer, reducing computational and memory burdens. Oh et al. (2021) updates only the feature extractor in the inner loop to encourage rapid movement of representations towards their corresponding classifiers. However, all these gradient-based approaches require backpropagation-based model training, which is computationally expensive and not well-suited to mobile devices. Metric-based methods learn to generate embeddings which facilitate simple classifier (Snell et al., 2017; Lee et al., 2019; Yang et al., 2021). Snell et al. (2017) learned a feature space where the representation of a given sample is close to the centroid of the corresponding class. Lee et al. (2019) leveraged a linear classifier, such as support vector machine (SVM), to gain better generalization with limited training data. Yang et al. (2021) made use of the statistics of the base classes to calibrate the distribution of the novel classes, enabling the generation of adequate samples for classifier training. However, a common model shared among all few-shot tasks may face bottlenecks when dealing with a wide diversity of tasks.

**Multi-domain Few-shot Learning.**  Multi-domain few-shot learning (FSL) considers situations where data come from multiple domains, which poses a significant challenge due to domain gaps. To address this issue, several works have proposed domain-dependent meta-models through task-adaptive embedding augmented with Feature-wise Linear Modulation (FiLM) layers (Oreshkin et al., 2018; Requeima et al., 2019; Bateni et al., 2020; Perez et al., 2018). Oreshkin et al. (2018) proposed a task encoding network to obtain a domain-dependent representation by shifting and scaling intermediate feature maps. Both Requeima et al. (2019) and Bateni et al. (2020) leveraged a few-shot adaptive classifier based on conditioned neural processes to capture domain-dependent information. Additionally, Vuorio et al. (2019) adopted domain-adaptive initialization, while Dvornik et al. (2020) proposed to build a multi-domain feature bank and select the most relevant features for the target task via gradient descent. Tseng et al. (2020) investigated out-of-domain FSL and proposed to improve model generalization by augmenting image features with feature-wise transformation layers. However, these methods either introduce auxiliary networks of large size or rely on gradient updates for model adaptation. Therefore, we propose the TANO module to extend metric-based FSL solutions.

**Normalization.**  Batch Normalization (BN) is a widely-used technique in deep learning (Ioffe & Szegedy, 2015). Empirical studies have shown that BN can accelerate the training of deep neural networks (DNNs) and make them less sensitive to optimizers and learning rates (Wu & He, 2018; Bjorck et al., 2018; Sun et al., 2020). Additionally, BN provides a regularization effect that helps prevent overfitting (Wu & Johnson, 2021). Furthermore, previous works have shown that BN layers can encode information about the data distribution

in the BN statistics. For example, Li et al. (2018) and Chang et al. (2019) have achieved effective domain adaptation by replacing the BN statistics learning in model training with those in testing. However, in few-shot learning (FSL) problems, calculating BN statistics using test data may lead to inaccurate estimation due to the limited data size. Moreover, when dealing with data from multiple domains during training, it is inappropriate to treat them as a single entity. Directly calculating the global BN statistics may lead to catastrophic performance. To address these challenges, other normalization methods, such as Instance Normalization (IN)(Ulyanov et al., 2016), Layer Normalization (LN)(Ba et al., 2016), and Group Normalization (GN)(Wu & He, 2018), have been proposed. IN applies normalization to a single image instead of an image batch, achieving great success in generative models. LN conducts a BN-like normalization only along the channel dimension and shows remarkable effectiveness in training sequential models (RNN/LSTM). GN is a variant of IN and LN, which is a more flexible sample-level normalization method by dividing channels into groups.

**Adaptive Network Architecture.** Recent works have demonstrated that selecting appropriate modules in neural networks can significantly benefit deep learning tasks (Rosenbaum et al., 2018; Veit & Belongie, 2018; Wang et al., 2018). Rosenbaum et al. (2018) introduced a router that recursively selects different function blocks for multi-task learning. This selection mechanism effectively reduces the interference between tasks. To reduce computational cost, Wang et al. (2018) and Veit & Belongie (2018) both develop a gate module to decide whether to skip some layers in neural networks. SkipNet integrates a reinforcement learning scheme to train the gate module. In this context, we introduce TANO to dynamically select BN statistics and normalization parameters. This selection ensures fast adaptation to specific domains with little computational overhead, while also leveraging the commonalities among different domains.

## 3 Preliminary and Problem Setting

### 3.1 Few-shot Learning via Meta-Learning

In a classification task, the set of instances and labels are denoted by $\mathcal{X}$ and $\mathcal{Y}$, respectively. An $N_w$-way $N_s$-shot problem refers to a classification task with $N_w$ categories and $N_s$ labeled examples in each category. The training data is represented as a support set $\mathcal{S} = \{(\mathcal{X}_s, \mathcal{Y}_s)\}$. Rather than training a machine learning model with a large support set, FSL considers the case where the number of shots in $\mathcal{S}$ is limited, such as when $N_s$ is as small as 1 or 5. Directly training a deep neural network over $\mathcal{S}$ is challenging as the model tends to over-fit.

Meta-learning is a promising approach to train effective classifiers with a limited support set, which involves two stages: meta-training and meta-testing. The goal of meta-learning is to handle few-shot tasks with *novel* classes, for which non-overlapping *base* classes are collected to simulate the tasks during meta-training. Specifically, a sequence of tasks, each with a support set $\mathcal{S} = \{(\mathcal{X}_s, \mathcal{Y}_s)\}$ and query set $\mathcal{Q} = \{(\mathcal{X}_q, \mathcal{Y}_q)\}$, are randomly sampled from the base classes during meta-training. A meta-model is optimized to minimize the average classification error over the sampled tasks, which predicts the query instances well conditioned on its corresponding support set. During the meta-testing stage, the learned meta-model generalizes to few-shot tasks with novel classes.

Metric-based meta-learning algorithms are commonly used in FSL problems (Snell et al., 2017; Vinyals et al., 2016; Sung et al., 2018), where a feature encoder $E(\cdot)$ and a metric function $M(\cdot)$ are learned jointly. The metric-based meta-model predicts the target input $\boldsymbol{x}_q$ based on its nearest support instance in the embedding space encoded by $E$ (which contains BN layers):

$$\hat{\boldsymbol{y}}_q = M(E(\mathcal{X}_s), \mathcal{Y}_s, E(\boldsymbol{x}_q)),$$
$$E^* = \arg\min_E \sum_{(\mathcal{S},\mathcal{Q})} \sum_{\substack{(\mathcal{X}_s, \mathcal{Y}_s), \\ (\boldsymbol{x}_q, \boldsymbol{y}_q) \in \mathcal{Q}}} L\left(\hat{\boldsymbol{y}}_q, \boldsymbol{y}_q\right). \tag{1}$$

Here, the loss function $L(\cdot)$ measures the prediction discrepancy between the metric's output and the ground truth. In metric-based meta-learning, all tasks share the same encoder $E$ and the metric function $M$, making it difficult to handle tasks where instances come from multiple domains, as shown in Table 2.

### 3.2 Problem Setting

The classical FSL paradigm assumes that each task is drawn from the same domain. However, in real-world scenarios, the tasks may exhibit heterogeneity in terms of data distributions due to various factors such as user preferences, device types, cultural differences, and regional variations. Thus, it is essential to extend FSL to handle tasks from multiple domains. To this end, we introduce multi-domain FSL, where each task can be sampled from different domains. As FSL consists of two stages, namely the meta-training stage and the meta-testing stage, multi-domain FSL can be further divided into two categories: intra-domain FSL and out-of-domain FSL, based on the domain differences between these two stages. We denote the set of all domains in Multi-domain FSL as $D_1, D_2, \cdots, D_R$, where $R$ is the total number of domains. We use $\mathcal{T}_{base}$ to represent the tasks used in the meta-training stage and $\mathcal{T}_{novel}$ to represent the tasks used in the meta-testing stage.

**Intra-domain Few-shot Learning.** Intra-domain FSL describes the problem of generalization within a group of related domains, where both the base classes and novel classes come from the same set of domains. This setting is illustrated in Figure 1, and can be formally defined as follows:

$$\mathcal{T}_{base}, \mathcal{T}_{novel} \subseteq \{D_1, D_2, \cdots, D_R\}. \tag{2}$$

Standard FSL can be considered as a special case of Intra-domain FSL, where the number of domains is equal to one.

**Out-of-domain Few-shot Learning.** Out-of-domain FSL refers to the cross-domain generalization problem, where the novel classes are sampled from domains that are unseen during the meta-training stage. This is described by Equation equation 3 and Figure 1.

$$\begin{aligned} \mathcal{T}_{base} &\subseteq \{D_1, D_2, \cdots, D_r\}, \\ \mathcal{T}_{novel} &\subseteq \{D_{r+1}, D_{r+2}, \cdots, D_R\}, 1 < r < R. \end{aligned} \tag{3}$$

Unlike in intra-domain FSL, the base classes and novel classes in out-of-domain FSL come from different domains. This poses a greater challenge for meta-learning as it requires the model to generalize to completely novel domains that were not seen during meta-training. This requires the model to learn domain-invariant features that can be used to generalize to unseen domains.

In addition, we assume that each few-shot task corresponds to a single domain. It is reasonable as the data of a particular user may not vary substantially across tasks. Furthermore, we assume that the domain index for a given task is not available during the meta-testing stage. Consequently, straightforward solutions such as independent domain-specific models may not be optimal without modeling the relationships between domains. Some recent works adopt domain-adaptive meta-learning to enable effective model adaptation with limited data. These methods require auxiliary networks of large size or rely on gradient updates, which may hinder their practical deployment on mobile devices. To address this limitation, we propose Task-aware Batch Normalization (TANO), a lightweight and plug-and-play FSL module for metric-based meta-learning algorithms.

## 4 Method

In this section, we explain the idea of TANO in the intra-domain FSL problem. Later, we show that TANO can also be easily extended to out-of-domain FSL.

### 4.1 TANO for Intra-domain Few-shot Learning

The diversity of data presents a challenge to assuming that instances from all tasks come from a single domain, particularly when dealing with users having diverse preferences that cannot be predicted by a single model. To address this issue, previous works (Li et al., 2018; Chang et al., 2019) have shown that Batch Normalization (BN) layers contain domain information, which is crucial when transferring domain

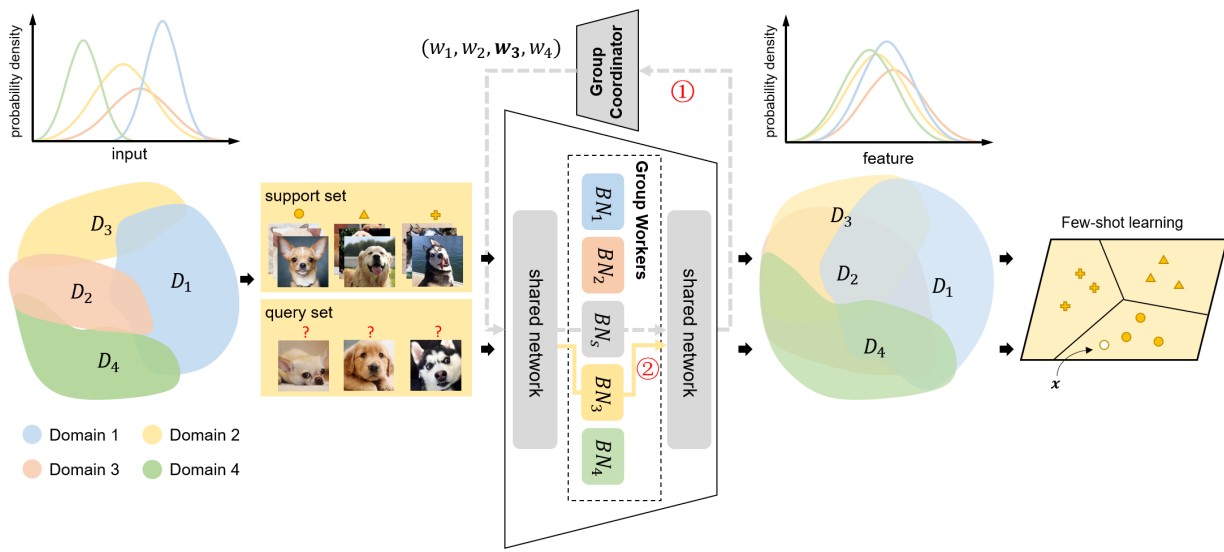

Figure 2: **Illustration of the proposed method.** In the multi-domain FSL problem, data from four different domains are referred as $D_1, \cdots, D_4$ with different colors. A FSL task is sampled from one domain. There are 5 Group Workers, Group Worker 1 to 4 correspond to the four domains and Group Worker $s$ corresponds to the global domain. $k$ is set to 1. The gray path denotes the first forward process via the global Group Worker and the Group Coordinator. The Group Coordinator identifies the domain weights $(w_1, \cdots, w_4)$ and provides it to the Group Workers. The yellow path denotes the second forward process via the third Group Worker, which is selected to be the encoder for this input by the Group Coordinator. The Group Workers narrow the domain gaps using different groups of BN statistics and achieve lightweight model adaptation. With proper BN statistics, the distribution differences of the intermediate features from different domains are significantly alleviated.

knowledge across domains. To handle the heterogeneity of multiple domains during meta-learning, we propose the TANO module consisting of two components: the Group Workers for the $R + 1$ groups of BN layers and the Group Coordinator for selecting the appropriate Group Worker to use (illustrated in Fig.2). The Group Workers pre-allocate multiple groups of BN parameters to capture domain-specific information, while the Group Coordinator dynamically selects the appropriate group-wise statistics for each task during the inference stage.

**Group Workers.** Each Group Worker includes the batch normalization (BN) parameters for all the BN layers in the network, including running mean, running variance, learnable weight and bias. Assuming the network has $J$ BN layers in total, each Group Worker contains $J$ groups of BN parameters. The first $R$ Group Workers correspond to the BN statistics of the $R$ domains, while the last one corresponds to the global BN statistics across all domains. The selection of one Group Worker corresponds to the allocation of its BN parameters into all BN layers in the network, which results in different feature embeddings.

Let $z$ be the intermediate feature in a deep neural network before a normalization layer, and assume that $z$ can be formulated as a 4D tensor indexed by $i = (i_N, i_C, i_H, i_W)$, where (N, C, H, W) correspond to the batch size, the number of channels, height and width of the tensor, respectively. If the $r$th group is selected and the feature is before the $j$th BN layer, $j \in [1, ..., J]$, then the feature $z$ is transformed by

$$\hat{z}_{i,j,r} = \frac{\gamma_{i,j,r}}{\sigma_{i,j,r}}(z_{i,j,r} - \mu_{i,j,r}) + \beta_{i,j,r}. \tag{4}$$

Here, $\gamma_{i,j,r}$ and $\beta_{i,j,r}$ are learnable weight and bias, respectively, $\mu_{i,j,r}$ is the mean and $\sigma_{i,j,r}$ is the variance, which are provided by

$$
\begin{aligned}
\mu_{i,j,r} &= \frac{1}{m} \sum_{u \in \mathcal{U}_i} z_{u,r,j}, \\
\sigma_{i,j,r} &= \sqrt{\frac{1}{m} \sum_{u \in \mathcal{U}_i} (z_{u,r,j} - \mu_{i,j,r})^2 + \epsilon},
\end{aligned}
\tag{5}
$$

where $\epsilon$ is a small constant, $\mathcal{U}_i = \{u | u_{\mathrm{C}} = i_{\mathrm{C}}\}$ is the set of pixels in which the mean and standard deviation are computed, and $m$ is the size of the set. Since the normalization layer in the corresponding Group Worker encodes the domain-specific information, the model processes domain-specific normalization for different tasks. In other words, multiple encoders are trained simultaneously, each of which characterizes one domain. For tasks from different domains, we can choose the corresponding encoder then we have a domain-invariant representation for all input data. This feature helps the model to adapt to a specific domain quickly and make full use of all input data to improve its performance on all domains.

**Group Coordinator.** Since the domain indices are unavailable in meta-testing stage, a Group Coordinator $\phi(\cdot)$ is further introduced to automatically identify the correct group of BN layers using the support set $\mathcal{S}$. We incorporate the global Group Worker (the last Group Worker) to get a universal embedding as the input of the Group Coordinator for instances from all domains. Let $E_r(\mathcal{S})$ be the embedding of $\mathcal{S}$ with the $r$th Group Worker, $r = 1, \cdots, R+1$. The objective of the Group Coordinator $\phi(\cdot)$ is to learn a mapping from $E_{R+1}(\mathcal{S})$'s to a probability distribution $\hat{\boldsymbol{w}} = (\hat{w}_1, \hat{w}_2, \cdots, \hat{w}_R) \in [0,1]^{1 \times R}$, where $\hat{w}_r$ is the probability estimation of $\mathcal{S}$ belonging to the $r$th domain. That is, $(\hat{w}_1, \hat{w}_2, \cdots, \hat{w}_R) = \phi(E_{R+1}(\mathcal{S}))$. Based on the prediction of the Group Coordinator, we can assign the Group Workers with high probability as the encoders for a given task. The selected Group Workers are updated in meta-training stage in the same way of BN layers and they are fixed in meta-testing stage.

In our implementation, the Group Coordinator consists of a series of fully-connected layers with ReLU as an activation function. A softmax layer is appended at the end. In this case, TANO only introduces several extra BN parameters and a Group Coordinator with a much smaller size compared to the widely used task-dependent modulation network for FSL domain adaptation (Perez et al., 2018; Vuorio et al., 2019; Oreshkin et al., 2018). Besides, TANO can adapt to a domain by selecting the correct Group Worker via the Group Coordinator instead of backpropagation.

**Implementation.** Incorporated with standard metric-based meta-learning algorithms such as Prototypical Networks, both the Group Workers and the Group Coordinator can be trained end-to-end via meta-learning. In the meta-training stage, we assume that a series of tasks $(\mathcal{S}, \mathcal{Q})$'s are sampled sequentially from different domains. Assuming the domain indices are known in the meta-training stage, we can use the one-hot encoding of the domain indices as the ground truth labels, denoted as $\boldsymbol{w}$, to guide the learning process of the Group Coordinator. The objective to learn the encoder $E(\cdot)$ and the Group Coordinator $\phi(\cdot)$ is provided by:

$$
(\phi^*, E^*) = \arg\min_{(\phi, E)} \sum_{r=1}^{R} v_r \sum_{(S_r, Q_r)} \sum_{\substack{(\mathcal{X}_s, \mathcal{Y}_s) \\ (\boldsymbol{x}_q, \boldsymbol{y}_q) \in Q_r}} [L(\hat{\boldsymbol{y}}_q, \boldsymbol{y}_q) + L(\hat{\boldsymbol{w}}, \boldsymbol{w})],
\tag{6}
$$

where $\hat{\boldsymbol{y}} = M(E(\mathcal{X}_s), \mathcal{Y}_s, E(\boldsymbol{x}_q))$ is the predicted image labels, $L(\cdot)$ is the cross-entropy loss, and $v_r, r = 1, \cdots, R$ is the weights of losses for data from different domains. By combining the two losses, the Group Worker and the Group Coordinator are learned simultaneously in an end-to-end manner. Through meta-learning, the Group Coordinator learns how to coordinate different Group Workers with BN layers across few-shot tasks.

Furthermore, our proposed method is extendable to scenarios where domain indices are not available in the meta-training stage. To address the issue of missing domain indices, we leverage the $k$-means clustering algorithm to cluster the training data and assign a pseudo domain label to each task. Utilizing the pseudo label can guide the Group Coordinator to determine the most appropriate domain for an incoming task.

Surprisingly, we observe that training with the pseudo label yields better performance than training with the ground truth domain indices, as demonstrated in Table 2. Consequently, we adopt the pseudo label to train our model for all settings, and consider the domain indices to be unavailable in both the meta-training and meta-testing stages.

### 4.2 TANO for Out-of-domain Few-shot Learning

The TANO module can be naturally extended to address out-of-domain FSL problems. Specifically, suppose that instances in the base classes are sampled from $R$ seen domains, denoted by $D_1, \cdots, D_R$, during the meta-training stage, while instances in the novel classes come from an unseen domain $D_{R+1}$. In this scenario, we can train a model with $D_1, \cdots, D_R$ using the algorithm introduced in Section 4.1. During the meta-testing stage, the Group Coordinator takes the support set $\mathcal{S}$ from the unseen domain $D_{R+1}$ as input and generates a corresponding domain prediction $\hat{\boldsymbol{w}}$. The Group Workers then perform normalization for the task data accordingly. Despite the unseen target domain during training, our experimental results demonstrate that TANO yields satisfactory performance in out-of-domain FSL (see Section 5.2).

In summary, we propose a lightweight module called TANO, which takes into account the multi-domain nature of FSL by pre-allocating multiple groups of BN parameters. By learning to combine these group-wise statistics, we integrate the TANO module with the encoder, which helps to find domain-invariant feature representations. TANO has two main advantages: first, it naturally accommodates multiple domains' properties during meta-training, and second, it incurs low computational cost in selecting groups during the inference stage.

### 4.3 Theoretical Analysis

Note that in Eqn.(4) and (5), if we treat the intermediate feature $z_{i,j,r}$ as a vector in $\mathbb{R}^m$ following Sun et al. (2020), the running mean $\mu_{i,j,r}$ and the running variance $\sigma_{i,j,r}$ actually map $z_{i,j,r}$ to a $(m-2)$-dimension sphere $\mathbb{S}^{m-2}_{\sqrt{m}}(\boldsymbol{0})$ with radius $\sqrt{m}$ and center origin via the mapping $(z_{i,j,r} - \mu_{i,j,r})/\sigma_{i,j,r}$. These two parameters represent the statistical approximation of data distribution. Moreover, the parameters $\gamma_{i,j,r}$ and $\beta_{i,j,r}$ further map the normalized $(z_{i,j,r} - \mu_{i,j,r})/\sigma_{i,j,r}$ to the sphere $\mathbb{S}^{m-2}_{\gamma_{i,r}\sqrt{m}}(\beta_{i,j,r})$, whose radius and center are determined by weight $\gamma_{i,j,r}$ and bias $\beta_{i,j,r}$, respectively. In the training stage, $\mu_{i,j,r}$ and $\sigma_{i,j,r}$ are calculated from the batch data, while $\gamma_{i,j,r}$'s and $\beta_{i,j,r}$'s are learned such that all $z_{i,r}$'s could be mapped to the sphere $\mathbb{S}^{m-2}_{\gamma_{i,j,r}\sqrt{m}}(\beta_{i,j,r})$. In the evaluation stage, the two parameters of the data distribution, i.e., $\mu_{i,j,r}$ and $\sigma_{i,j,r}$, are replaced by their moving averages in the training stage. Together with the learned $\gamma_{i,j,r}$ and $\beta_{i,j,r}$, these parameters aim to map the test vectors to the same sphere as that in training. If not, the prediction results would be untraceable (see Table 3 for some evidence). That may be because other network parameters except the normalization layers are learned to encode the inputs based on the specific normalization used in training. Therefore, appropriate normalization statistics of data distribution play an important role in evaluation. Note that the above-mentioned parameters are domain-specific. Inputs from different domains lay on different spheres. The TANO module provides a more appropriate normalization and thus achieves better accuracy. In addition, with the TANO module, inputs from different domains tend to be normalized to a common sphere. As a result, network parameters except for the BN layers would access more diverse samples with proper normalization compared with training a model assuming the data come from a single domain.

## 5 Experiments

### 5.1 Intra-domain Few-shot Learning

First, we evaluate our proposed method when dealing with data from multiple domains *without domain indices*. To create intra-domain few-shot classification tasks, we combine multiple widely used datasets including *mini*ImageNet (Ravi & Larochelle, 2017), CUB (Welinder et al., 2010), Dogs (Khosla et al., 2011), Cars (Krause et al., 2013) to form a meta dataset, following the train/set splits used in previous works. Details of the datasets can be found in Table 1.

Table 1: **Summarization of the datasets.** We additonally collect and split CUB, Cars and Dogs.

| Datasets | $mini$ImageNet | CUB | Cars | Dogs |
|---|---|---|---|---|
| Training | 64 | 100 | 98 | 60 |
| Validation | 16 | 50 | 49 | 30 |
| Testing | 20 | 50 | 48 | 30 |
| Split | Ravi & Larochelle (2017) | random | random | random |

Table 2: **Intra-domain few-shot classification results trained on $mini$ImageNet, CUB, Cars and Dogs.** The models are evaluated using 5-way 1-shot and 5-way 5-shot tasks. *For a fair comparison, we implement SUR based on the same pre-trianed weights as TANO, which achieves lower accuracy than the ImageNet pre-trained model in Dvornik et al. (2020).

| | Backbone | $mini$ | CUB | Cars | Dogs | mean | $mini$ | CUB | Cars | Dogs | mean |
|---|---|---|---|---|---|---|---|---|---|---|---|
| | | | | 1 shot | | | | | 5 shot | | |
| MultiModels | Conv | 48.29 | 63.14 | 46.67 | 52.99 | 52.52 | 67.60 | 80.99 | 67.94 | 71.82 | 72.11 |
| ComModel | Conv | 48.99 | 59.94 | 41.61 | 50.82 | 50.34 | 80.19 | 58.87 | 70.29 | 69.45 | 69.70 |
| AdaBN | Conv | 49.98 | 59.38 | 42.67 | 49.51 | 50.39 | 69.22 | 79.70 | 60.49 | 69.55 | 69.74 |
| MMAML | Conv | 46.76 | 62.76 | 44.75 | 50.44 | 51.17 | 62.48 | 78.44 | 62.39 | 66.00 | 67.33 |
| **TANO** | Conv | 49.71 | 60.60 | 48.22 | 52.51 | 52.76 | 70.24 | 80.97 | 68.43 | 72.89 | 73.13 |
| MultiModels | Res18 | 60.86 | 76.21 | 75.97 | 75.31 | 72.09 | 76.90 | 88.18 | 89.04 | 87.56 | 85.42 |
| ComModel | Res18 | 61.05 | 76.60 | 71.35 | 74.52 | 70.88 | 79.49 | 88.70 | 87.76 | 88.24 | 86.05 |
| SUR* | Res18 | 58.05 | 73.28 | 64.10 | 65.86 | 65.32 | 74.07 | 86.65 | 82.37 | 82.37 | 81.37 |
| **TANO** | Res18 | 62.68 | 77.47 | 74.68 | 75.97 | 72.70 | 79.63 | 88.88 | 88.36 | 89.04 | 86.48 |

**Baselines.** We compare our method with several baselines, as introduced as follows. (1) **MultiModels.** One basic idea to deal with intra-domain FSL problems is to train multiple models for different domains separately. This training paradigm is denoted as MultiModels. (2) **ComModel.** Alternatively, one may mix the training data from different domains and train a common model for all domains. This training paradigm is referred to ComModel. (3) **AdaBN.** We adopt AdaBN (Li et al., 2018) as a baseline based on our trained ComModel, which adapt the running-mean and running-variance of the batch normalization layer to the mean and variance of target domain. (4) **MMAML.** We also include MMAML (Vuorio et al., 2019) as a baseline, which leverages a task encoder network to modulate the model for each task with a corresponding domain index. (5) **SUR.** SUR (Dvornik et al., 2020) involves training a set of semantically different feature extractors to obtain a multi-domain representation.

**Implementation.** We implement the above methods with PrototNets using four-layer ConvNet (Vuorio et al., 2019) and ResNet-18 (Chen et al., 2019) backbones. We implement MMAML with the 4-layer ConvNet backbone only, as the paper did not introduce the implementation on ResNet backbones. Following Rusu et al. (2019), we pr-etrain the backbones to jointly classify all classes in meta-training set using cross-entropy loss. For ProtoNets, we use Euclidean distance as the similarity metric. In meta-training stage, we sample 5-way 1-shot tasks. The optimizer is SGD with a initial learning rate 0.001. We train for 200 epochs and use cosine annealing for learning rate decay. All the algorithms are tested on 600 trials of 5-way 1-shot and 5-way 5-shot tasks. The number of samples in a query set is 15.

**Results.** 1. **Comparison with ComModel and AdaBN.** As can be seen in Table 2, TANO outperforms ComModel across different domains and backbones. Moreover, AdaBN exhibits similar performance to ComModel, suggesting that merely adapting the running-mean and running-variance does not lead to improved performance in the intra-domain FSL scenario. It is worth noting that MultiModels achieves higher classification accuracy than ComModel in most cases, which contradicts the expectation that more data would result in better performance. This finding suggests that domain gaps in the meta-training data could

Table 3: **MultiModels across domains for 5-way 1-shot and 5-shot tasks.** We compare TANO with the four models trained on *mini*ImageNet, CUB, Cars, and Dogs separately across different domains. In each column, we train a model on one specific domain and evaluate the performance on four domains respectively. The values outside bracket are the performance of MultiModels, while the values inside the bracket are the performance improvement of TANO.

| Test/Train | mini | CUB | Cars | Dogs |
|---|---|---|---|---|
| | | 1 shot | | |
| mini | 60.76(+1.92) | 35.83(+26.85) | 34.47(+28.21) | 35.95(+26.73) |
| CUB | 45.83(+30.64) | 76.33(+1.14) | **35.99(+41.48)** | 37.39(+40.08) |
| Cars | **32.23(+42.45)** | 36.22(+38.46) | 75.54(-0.86) | **31.22(+43.46)** |
| Dogs | 54.66(+21.31) | 41.88(+34.09) | **30.41(+45.56)** | 74.48(+1.49) |
| | | 5 shot | | |
| mini | 76.90(+3.15) | 51.76(+28.29) | 48.14(+28.26) | 54.69(+25.46) |
| CUB | 62.78(+26.47) | 88.18(+1.07) | **48.20(+41.05)** | 56.59(+32.66) |
| Cars | **44.21(+43.39)** | 46.03(+41.57) | 89.04(-1.44) | **44.70(+42.90)** |
| Dogs | 69.19(+18.37) | 57.21(+30.35) | **40.96(+46.60)** | 86.92(+0.64) |

Table 4: **Intra-domain few-shot classification test time and number of parameters.**

| | Backbone | 1-shot(/ms) | 5-shot(/ms) | #Params(M) |
|---|---|---|---|---|
| ComModel | Conv | 3.31 | 4.09 | 0.13 |
| MMAML | Conv | 377.35 | 448.33 | 0.89 |
| **TANO** | Conv | 9.08 | 11.01 | 0.13 |
| ComModel | Res18 | 6.94 | 13.64 | 10.70 |
| SUR | Res18 | 147.85 | 159.55 | 10.87 |
| **TANO** | Res18 | 21.29 | 41.02 | 10.78 |

negatively impact the model's performance. With the introduction of only a few additional parameters, TANO successfully closes the domain gaps and achieves improved classification accuracy.

2. **Comparison with MultiModels.** In comparison with MultiModels, TANO obtains competitive results with much lower memory usage. MultiModels requires four different models to handle data from four domains, which leads to higher memory consumption. To test the generalization capacity, we evaluate the models on domains beyond their seen domains. As depicted in Table 3, a significant performance drop is observed when a single model is tested on other domains, indicating its limited generalization ability. In contrast, TANO achieves better performance than a single model across different domains. Additionally, for datasets like *mini*ImageNet, TANO outperforms a single model trained solely on *mini*ImageNet. The reason behind this could be that, while the BN layers are domain-specific, the rest of the backbone is shared. Therefore, training on data from different domains helps to learn better network weights, resulting in more powerful embeddings.

3. **Comparison with MMAML and SUR**. TANO demonstrates superior performance compared with MMAML (Vuorio et al., 2019) and SUR (Dvornik et al., 2020). MMAML involves backpropagation to achieve model adaptation, which can be computationally and memory intensive, particularly for mobile devices. On the other hand, SUR uses FiLM layers to capture domain-specific information without introducing many additional parameters, but gradient is also employed in the feature selection scheme. In contrast, TANO efficiently adapts to a domain by automatically selecting the corresponding normalization groups, without relying on backpropagation. We conducted experiments to compare the average running time of each task for MMAML and TANO over 600 trials on a single V100. As indicated in Table 4, the average running time of TANO is nearly 40 times shorter than MMAML, and several times shorter than SUR, demonstrating its significant time efficiency advantage. Moreover, the extra parameters introduced by TANO are negligible

Table 5: **Out-of-domain few-shot classification results.** We use the leave-one-out setting to select the test domain and use the remaining as the training domains. Models are evaluated on 5-way 1-shot and 5-way 5-shot tasks.

| Domain\Model | | LFT | TANO |
|:---:|:---:|:---:|:---:|
| 1 shot | CUB | $43.77 \pm 1.32$ | $\mathbf{45.44 \pm 0.77}$ |
| | Cars | $35.12 \pm 1.54$ | $\mathbf{35.14 \pm 0.68}$ |
| | Dogs | $36.91 \pm 1.29$ | $\mathbf{52.90 \pm 0.81}$ |
| 5 shot | CUB | $57.53 \pm 1.23$ | $\mathbf{59.00 \pm 0.75}$ |
| | Cars | $43.64 \pm 1.44$ | $\mathbf{44.81 \pm 0.72}$ |
| | Dogs | $50.62 \pm 1.32$ | $\mathbf{65.35 \pm 0.74}$ |

compared with the size of backbone, and much less than MMAML, demonstrating TANO's lightweight design for multi-domain FSL. Additionally, TANO eliminates the need for domain indices during meta-training stage, which could reduce the cost of data annotation. In summary, TANO is more efficient and lightweight compared with MMAML.

### 5.2 Out-of-domain Few-shot Learning

We also evaluate the performance of TANO for Out-of-domain FSL, and compare it with LFT (Tseng et al., 2020), a recently proposed FSL method to address this problem. The core idea of LFT is to uses feature-wise transformation layers to simulate different feature distributions and a learning-to-learn approach to optimize the hyper-parameters of these layers to capture the variations of image feature distributions across different domains.

**Implementation.**   Following LFT, we incorporate TANO with MatchingNet. We adopt the leave-one-out setting by selecting an unseen domain from CUB, Cars, or Dogs domains. The *mini*ImageNet and the remaining domains then serve as the seen domains for training. After the meta-training stage, we evaluate the trained model on the selected unseen domain. We employ the pre-trained ResNet-18 as the backbone of MatchingNet.

**Results.**   As shown in Table 5, TANO significantly outperforms LFT in all domains. Note that the target domain is unseen in meta-training stage. LFT proposes to simulate the distribution of the unseen domains by introducing some noise on BN layers' learnable weight and bias. The introduction of the noise in the meta-training stage can benefit the robustness of the model. However, the BN statistics are fixed after meta-training stage. Thus the model can not provide an accurate domain-specific estimation of the BN statistics for the target unseen domain. By providing multiple groups of BN parameters and a Group Coordinator, TANO allows us to use the combinations of BN parameters from seen domains to estimate that of the unseen domains. This selection and merge scheme enables the model to predict the BN information of the target domain more flexibly and precisely. As mentioned in AdaBN (Li et al., 2018), an accurate estimation of BN statistics is essential in cross-domain learning tasks, which explains the advantage of TANO in out-of-domain FSL.

### 5.3 Standard Few-shot Learning Benchmark

In this section, we adapt TANO into the standard FSL problem. We incorporate TANO with ProtoNets using ResNet-12 backbone (Lee et al., 2019). We consider several baseline methods including TADAM (Oreshkin et al., 2018), DC (Lifchitz et al., 2019), MCRNet (Zhong et al., 2021), MetaOptNet (Lee et al., 2019), Shot-Free (Ravichandran et al., 2019), MTL (Sun et al., 2019), and TapNet (Yoon et al., 2019). As shown in Table 6, TANO improves the ProtoNet (vanilla) remarkably, and achieves SOTA results compared with the baseline methods on *mini*ImageNet and *tiered*ImageNet.

Table 6: **Standard few-shot classification results.** We conduct 5-way 1-shot/5-shot experiments on miniImageNet and tieredImageNet. TANO is achieved with ProtoNets. The results are the average accuracies with 95% confidence intervals.

| Model | *mini*Imagenet | | *tiered*Imagenet | |
|---|---|---|---|---|
| | 1 shot | 5 shot | 1 shot | 5 shot |
| TADAM (Oreshkin et al., 2018) | $58.50 \pm 0.30$ | $76.70 \pm 0.30$ | - | - |
| FEAT(Ye et al., 2020) | $62.96 \pm 0.02$ | $78.49 \pm 0.02$ | - | - |
| DC (Lifchitz et al., 2019) | $62.53 \pm 0.19$ | $78.95 \pm 0.13$ | - | - |
| MCRNet (Zhong et al., 2021) | $62.53 \pm 0.19$ | $79.77 \pm 0.19$ | - | - |
| MetaOptNet (Lee et al., 2019) | $62.64 \pm 0.61$ | $78.63 \pm 0.46$ | $65.99 \pm 0.72$ | $81.56 \pm 0.53$ |
| Shot-Free (Ravichandran et al., 2019) | $59.04 \pm 0.43$ | $77.64 \pm 0.39$ | $66.87 \pm 0.43$ | $82.64 \pm 0.39$ |
| MTL (Sun et al., 2019) | $61.20 \pm 1.80$ | $75.53 \pm 0.80$ | $65.62 \pm 1.80$ | $80.61 \pm 0.90$ |
| TapNet (Yoon et al., 2019) | $61.65 \pm 0.15$ | $76.36 \pm 0.10$ | $63.08 \pm 0.15$ | $80.26 \pm 0.12$ |
| ProtoNet(Snell et al., 2017) (vanilla) | $60.37 \pm 0.83$ | $78.02 \pm 0.57$ | $65.65 \pm 0.92$ | $83.40 \pm 0.65$ |
| TANO | $63.00 \pm 0.89$ | $79.08 \pm 0.59$ | $66.58 \pm 0.94$ | $84.07 \pm 0.62$ |

Table 7: **On-device evaluation of latency and energy.** We test the latency and energy cost of different methods to inference 100 times for 1-shot tasks.

| methods | latency(s) | energy cost(J) |
|---|---|---|
| SUR | 352.66 | 2458.89 |
| LFT | 9.93 | 61.57 |
| TANO | 7.75 | 39.54 |

### 5.4 Evaluation on Edge-Device

In this section, we evaluate the latency and energy cost of TANO on NVIDIA Jetson Nano (Cass, 2020), which is a powerful embedded platform that can be used as an edge device for running deep learning models in real-time. With its small form factor and low power consumption, the Jetson Nano is an ideal platform for deploying intelligent edge devices that require high computational power while being energy-efficient. We compare the proposed method with SUR (Dvornik et al., 2020) and LFT (Tseng et al., 2020). To evaluate the performance of the proposed methods, we conducted 100 inferences for a 1-shot task on the Jetson Nano platform while recording the corresponding latency and energy cost. To account for potential variations in the hardware conditions, we repeated the experiment 5 times and computed the average latency and energy cost as the final results. As shown in Table 7, TANO remarkably enhances the hardware efficiency in comparison to SUR. In the case of LFT, TANO necessitates less latency and energy cost, while also providing a considerably superior performance.

## 6 Conclusion

In this paper, we have proposed a novel lightweight module called TANO for intra-domain and out-of-domain FSL problems. By utilizing domain-specific BN layers and a coordinator to select the appropriate BN layer for inputs from different domains, our method effectively addresses domain gaps in Few-shot Learning. Furthermore, TANO can be extended to out-of-domain FSL tasks by estimating appropriate BN statistics for unseen domains, and can also improve the performance of single domain standard FSL problems by using dummy domains. Our extensive experiments have demonstrated that TANO achieves faster inference, smaller model sizes, and stronger generalization compared with existing meta-learning methods. TANO provides a promising solution for Few-shot Learning in real-world scenarios, where multiple domains and limited data are common challenges.

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
