# OpenReview forum: "Out-of-distribution Few-shot Learning For Edge Devices without Model Fine-tuning"
_TMLR — Withdrawn by Authors_

### Review · Reviewer_1cWR · 2023-04-19

**Summary Of Contributions:**

This paper proposes TANO, an approach for light-weight few-shot learning, intended for use in edge devices with limited computation capabilities. They target in particular the ‘out-of-distribution’ scenario where the tasks appearing at meta-test time originate from different domains/datasets compared to those seen at meta-training time, inducing a distribution shift. They propose to address this problem by learning a collection of batch normalization statistics and parameters: one for each training domain/dataset that is available, and an additional ‘global’ one. They also meta-train a coordinator that selects an appropriate member of that collection to use given a new task, allowing to tackle new tasks at test time without gradient descent adaptation. In particular, the global set is first used to embed the support set of each task, and this global embedding is then fed to the group coordinator which then chooses a member out of the collection of R sets of BN layers for the particular task, and the forward pass is repeated with the chosen BN layers. The authors show that on some datasets, TANO performs better sometimes compared to some relatively weak baselines.

**Audience:**

Yes

**Broader Impact Concerns:**

I have no broader impact concerns about this paper.

**Claims And Evidence:**

No

**Requested Changes:**

- Cite, discuss and compare experimentally to the relevant baselines for this topic (see comment above in Weaknesses section and the references)

- For SUR, compare against the full variant as well as the parametric family variant (see comment above)

- [Clarity / correctness] In the text above Figure 1, terminology is used incorrectly. For instance, ‘the support set consists of data from multiple distributions, and the query set contains data from the seen domains with unseen classes’ – this sentence uses the words ‘support’ and ‘query’ incorrectly. They should instead be ‘meta-train’ and ‘meta-test’ (or more simply train and test). All this terminology should be carefully defined. In the literature, support and query are used to refer to the *within-task* training and test sets that are used for task-specific adaptation and task-specific evaluation of the general model, whereas meta-training and meta-testing refer to the data across tasks (and domains) that is available overall at (meta-)training time to (meta-)learn a general model.

- [Correctness / related work] The authors repeat in a few places that previous methods “either introduce auxiliary networks of large size or rely on gradient updates for adaptation”, in order to motivate TANO. However, it’s a bit odd then that TANO also introduces an auxiliary network. If the authors want to substantiate the argument that TANO’s auxiliary network is much smaller than that of previous works, more emphasis should be given to this point (e.g. by describing the architecture). Several previous works like CNAPs (Requeima et al, which is cited in the paper) and FLUTE (see references above), for instance, already use a relatively lightweight auxiliary network (5 conv layers). Note also that CNAPs performs no gradient updates to the deep model either at inference time, as it’s fully amortized (only a forward pass through the auxiliary network is required in order to produce the task-specific parameters). So it seems that this placement within the literature here isn’t really accurate.

- [soundness, unsubstantiated claim/hypothesis] The authors say that ‘directly calculating the global BN statistics may lead to catastrophic performance’. It would be great if the authors can actually show the results of this experiment. Actually showing catastrophic performance there would better motivate TANO. As it stands, it’s only a hypothesis. Note as well that several few-shot learning works (both for intra- and cross- domain) put batch norm to train mode at meta-test time, thus calculating statistics from the support set of each test task, rather than re-using pre-calculated statistics from meta-training. This works quite well in practice (see the analysis in the TaskNorm paper, for example), so the authors need to compare to this variant as well.

- [soundness, technical strength] The paper has no ablations! An important one, for instance, is about investigating whether one needs to maintain a collection of both BN parameters and BN statistics. For instance, FLUTE has a simpler version that maintains only the former. For statistics, then, one can try different variants: use global statistics from meta-training time, or set BN to train mode at test time which means using support set statistics from each task, or doing something akin to TaskNorm. These variants needs to be examined to justify the proposed design choice.

- [clarity] I’m a bit confused what the index i is in each of the gamma, beta, mu and sigma in Equation 4. These should be indexed by r (the index in the collection), j (indicating which BN layer) and the channel index, right? Above Equation 4, i is defined as a 4D tensor which is confusing. Also, below Equation 5, U_i is defined as a set of pixels. This isn’t generally true, right? Only in the first layer could this be a set of pixels. Am I missing something?

- [clarity] In the section explaining the Group Coordinator, the authors say that E_r(S) is the embedding of the support set S w.r.t group worker r. But S is a set, not a single example. How, then, is the embedding of a set computed? I’m assuming an average is taken, or some other aggregation? Need to specify.

- [clarity] Does the Group Coordinator do a hard or soft selection? It would be great to ablate both options.

- [clarity, technical correctness] How are weights v_r set? Additionally, how is hyperparameter tuning done generally in the paper? This is very important to explain for settings like cross-domain classification, as the procedure used there can largely affect conclusions. What was the validation set? Held-out classes of base domains? Held-out domains? Please clarify.

- [experiments] Results are quite weak compared to baselines, even though the baselines considered here aren’t SOTA. Why aren’t confidence intervals reported in Table 2? Should do this and bold accordingly. It seems that several methods outperform TANO in several places there. In Table 3, confidence intervals are reported but bolding isn’t done correctly. For instance, LFT (which is the only baseline shown there) performs equally well as TANO in several cases and yet isn’t bolded.

- [clarity, correctness] In Table 4, how come the number of parameters is the same for ComModel as for TANO? TANO has additional parameters, like the Group Coordinator, and more than one sets of BN parameters, right? This doesn’t seem correct.

- [minor, clarity] The problem setting in section 3.2 isn’t precise enough. For example, Equation 2 doesn’t indicate in any way that the classes are disjoint between those in ‘base’ and those in ‘novel’

- [minor, notation] it’s odd that M (in Equation 1) takes a label as input, because M denotes a metric function, which typically takes as input two objects and outputs their similarity (regardless of labels).


**Strengths And Weaknesses:**

Strengths
========
- The paper addresses a relevant and challenging problem: few-shot learning on edge devices, which introduces memory and computation constraints in addition to the usual challenges of few-shot learning (effectively updating a model without access to a lot of data for a new task). Further, they explore OOD variants of the problem too, where new tasks appearing at (meta-) test time are substantially different from those seen during (meta-)training time.



Weaknesses
===========
- The experimental evaluation is limited to 4 simple datasets that aren’t too semantically different from each other, making even the OOD scenarios easier than they might be in practice. Several related works have tackled cross-domain few-shot learning recently on more challenging benchmarks like Meta-Dataset [1], VTAB [2] and the combined Meta-Dataset+VTAB [3], for example, which would be more appropriate benchmarks for this study, as the datasets/tasks used there contain significantly larger variety (natural images, synthetic, textures, medical images, etc)

- The authors don’t compare TANO against the appropriate baselines (see next point - there is a plethora of related work on this topic). Therefore I am not able to assess how well this model performs relative to the existing state of the art.

- Related work: The authors claim that ‘lightweight adaptation remains a challenge’, but there is a plethora of recent related work that addresses this, which is unfortunately not cited by this paper. For example: URT [4] is a related approach to SUR (which the authors cite), but alleviates the need for gradient descent at inference time (they instead have a meta-training phase that meta-learns how to select relevant features from the universal representation, given the specification of a task). Similarly, FLUTE [5] is a method that enables light-weight adaptation (at test time, they do perform some gradient descent, but only on FiLM layers, which is very minimal, and their approach also works well with 0 steps of gradient descent). FLUTE in particular is very similar to TANO in terms of the proposed method: they train a collection of feature extractors at training time that differ only in their BN parameters (all other parameters shared) and train a network to pick the most appropriate set out of this collection for every new task. Other related methods are Tri-M [6], URL [7], TSA [8] and FiT [9], all of which are specifically designed to be light-weight but very effective at few-shot learning, and have been tested on OOD few-shot learning scenarios too, on more challenging benchmarks than the one considered here. See references below. The similarities and differences between TANO and each of these methods should be discussed and TANO should be compared against them experimentally. Finally, regarding the difficulty of computing batch normalization statistics from few examples in a new task, as well as the difficulty of reusing global BN stats for new tasks at test time, the authors should cite and compare against TaskNorm [10], which is designed to address exactly this problem and has done extensive comparisons of different flavors of BN for few-shot learning.

- The authors seem to have compared not against full SUR, but its ‘parametric family variant’. Instead, should compare against both (the parametric family variant underperforms the full variant in several situations) for a reliable comparison against TANO.

- The paper has several clarity issues and missing information that prevents from fully understanding the details of TANO (see below in Requested changes for more details)

- The experimental investigation is quite weak. There are no ablations made to justify the design choices of TANO. Comparison is done only against a small set of baselines that aren’t SOTA, and even then, TANO is often outperformed by them (and the authors aren’t very transparent about this, e.g. in the way that the text is written and the bolding (or lack thereof) in the tables).


References
==========
- [1] Meta-Dataset: A Dataset of Datasets for Learning to Learn from Few Examples. Triantafillou et al. ICLR 2020.

- [2] A Large-scale Study of Representation Learning with the Visual Task Adaptation Benchmark. Zhai et al. 2020.

- [3] Comparing Transfer and Meta Learning Approaches on a Unified Few-Shot Classification Benchmark. Dumoulin et al. NeurIPS 2022.

- [4] A Universal Representation Transformer Layer for Few-Shot Image Classification. Liu et al. ICLR 2021.

- [5] Learning a Universal Template for Few-shot Dataset Generalization. Triantafillou et al. ICML 2021.

- [6] A Multi-Mode Modulator for Multi-Domain Few-Shot Classification. Liu et al. ICCV 2021.

- [7] Universal Representation Learning from Multiple Domains for Few-shot Classification. Li et al. 2021.

- [8] Cross-domain Few-shot Learning with Task-specific Adapters. Li et al. 2021

- [9] FiT: Parameter Efficient Few-shot Transfer Learning for Personalized and Federated Image Classification. Shysheya et al. ICLR 2023.

- [10] TASKNORM: Rethinking Batch Normalization for Meta-Learning. Bronskill et al. ICML 2020.

---

### Review · Reviewer_8LTr · 2023-05-18

**Summary Of Contributions:**

This paper proposes a simple technique for few-shot learning by leveraging task specific batch normalization parameters in a multi-domain, multi-task meta-learning setting.

**Audience:**

Yes

**Claims And Evidence:**

No

**Requested Changes:**

 - I advise changing the title by removing "For Edge Devices" and perhaps adding "Lightweight". It would better capture the essence of the method and the emphasis of the paper
 - I advise changing "Standard FSL" to "single-domain FSL" to minimize confusion
 - It's not made clear what the difference between "domain" and "task" is
 - It's also not clear whether "out-of-distribution" and "out-of-domain" mean the same thing
 - End of Page 4, "as shown in Table 2" seems like a mistake in reference. If not, please explain how Table 2 supports such a claim
 - Figure 2 caption, "k is set to 1" -- what is k?
 - Table 1 is very unclear to the point of useless. What are the numbers in the table?
 - Table 2: again, what are the "classification result" numbers? Are they accuracies? Errors?
 - Table 3: "values inside the bracket are the performance improvement of TANO" -- do you mean in the case of "32.23(+42.45)", TANO performance is 74.68? The numbers don't match with what's in Table 2
 - "In comparison with MultiModels, TANO obtains competitive results with much lower memory usage." Please provide evidence


**Strengths And Weaknesses:**

Strengths
 - simple yet effective idea
 - relevant topic to TMLR
 - Figures 1 and 2 are well illustrated and easy to follow
 - comprehensive experiment results

Weaknesses
 - the "For Edge Devices" claim in the paper title is unsubstantiated. Only the final subsection touched upon edge devices with insufficient experiments
 - comparison with fine-tuning methods is not sufficient
 - Tables are often unclear and lacking detailed information
 - no ablation studies to support the validity of the method

---

### Review · Reviewer_EUtw · 2023-06-11

**Summary Of Contributions:**

The paper
1. addresses the problem of out-of-distribution (OOD) adaptation on the device.
2. proposes a method called task-aware normalization to perform the adaptation without finetuning model weights.
3. performs experiments to validate the idea.

To summarize my recommendation, I think the paper has some value to the community but the contribution in novelty is low from my perspective. I think the paper could be a good workshop paper instead of a full research paper.

**Audience:**

Yes

**Broader Impact Concerns:**

The batchnorm statistics on device may reveal some information about the end users. Potential risk may happen here. Can you discuss around this question?

**Claims And Evidence:**

Yes

**Requested Changes:**

Grammar issues:
* p2: "One practical example for is the face identification in ..."
* discuss relatedness and differentiation from federated learning.

Please try to address the questions listed in weakness.

**Strengths And Weaknesses:**

Strength:
1. The topic is useful to real device applications.
2. The proposed approach seems to work well, although it does not achieve SOTA.
3. The proposed method is simple to implement on any device.

Weakness:
1. The novelty is weak as a TMLR submission. Batch norm is a well known adjustment for distribution shift. It often needs to be adjusted in the eval distribution.
2. Since the paper discusses on-device problem, federated learning needs to be discussed and differentiate. e.g., Dong et al. Hyperspherical Federated Learning 2022 proposed similar idea on normalizing for distribution issues across devices.
3. The experimental validation is reasonable but the approach does not perform significantly better than existing approaches.
4. Given the fact the contribution is limited to batchnorm updates only and limitation in experiments understanding, I think the paper will be more suitable to a workshop paper instead of a full paper.
5. Since the paper is about few shot learning, one can assume the examples on each device is relatively low. Then how would the batchnorm statistics be accurate in that situation?

---

> ### Author Response · Authors · 2023-06-25
> **Response to Reviewer EUtw**
>
> **Q1 and Q4: Novelty.**
>
> Although Batch Normalization is a well-known adjustment for distribution shift, it's not easy to accurately estimate the batch norm statistics with very limited data in a few-shot learning setting. In this work, we propose using group workers to estimate a set of batch norm statistics during training, while selecting one of them during testing. In this case, we can achieve accurate estimation of batch norm statistics compared to existing works in few-shot learning.
> The proposed method is a lightweight module for metric-based meta-learning algorithms that significantly
> boost model performance in FSL with data from multiple domains without gradient updates.
>
> **Q2: Comparison with Dong et al. (2022).**
>
> First and foremost, we consider a different setting from that in Dong et al. (2022). In our setting, we focus on scenarios where gradient descent is not permitted on edge devices, which necessitates the use of gradient-free adaptation for non-IID data. Conversely, Dong et al. (2022) considers a standard federated learning setting where each client updates the model locally and synchronizes with the server. In terms of method, Dong et al. (2022) constructs an orthonormal classifier and learns feature extractors for each client that conform to this common classifier, in order to address the non-IID issue across different clients. In contrast, we employ group workers to track domain information, along with a group coordinator that learns to identify the corresponding batch norm statistics, thus mitigating distribution shift.
>
> **Q3: Performance.**
>
> In the experiment, we compared our proposed method with few-shot learning methods, including MMAML, SUR, LFT, among others. Our method outperformed these methods and also had better latency and computational costs when evaluated on devices.
>
> **Q5: Accurate estimation of batchnorm statistics.**
>
> While Batch Normalization is a well-known adjustment for distribution shift, it can be challenging to accurately estimate the batch norm statistics with limited data in a few-shot learning setting. In response to this challenge, our work introduces a new approach. We propose the use of group workers to estimate a set of batch norm statistics during training. Then, during testing, we select one of these statistics. This method allows us to achieve a more accurate estimation of batch norm statistics compared to other existing few-shot learning works.
>
> Importantly, during the training phase, our model is trained on the server, which has access to abundant data from various domains. This includes many few-shot tasks from each domain. Consequently, the group workers can accurately aggregate the batch norm statistics for each domain. Moreover, the group coordinator can be effectively trained to predict the domain based on just a few samples.
>
> Once trained, we can deploy this model on edge devices. Given the approach we've taken during the training phase, the distribution calibration based on batch norm statistics is likely to be accurate even when the number of examples on each device is relatively low. This method, therefore, addresses the central challenge in few-shot learning scenarios.
>
>
> [1] Dong et al. Hyperspherical Federated Learning, ECCV'22

---

### Note · Authors · 2023-06-25

I have read and agree with the venue's withdrawal policy on behalf of myself and my co-authors.